# Impact of Parental Food Choices on Nutritional and Metabolic Status of Children with Type 1 Diabetes

**DOI:** 10.3390/foods12101969

**Published:** 2023-05-12

**Authors:** Claudia López-Morago, Jesús Domínguez-Riscart, Ana García-Zarzuela, Alfonso María Lechuga-Sancho

**Affiliations:** 1Biomedical Research and Innovation Institute of Cadiz (INIBiCA), 11009 Cadiz, Spain; claudia.lopezmorago@gmail.com (C.L.-M.); anag.zarzuela@gmail.com (A.G.-Z.); alfonso.lechuga@uca.edu.es (A.M.L.-S.); 2Pediatric Endocrinology Unit, Pediatric Department, Puerta del Mar University Hospital, 11009 Cadiz, Spain; 3Mother and Child Health and Radiology Department, Faculty of Medicine, Cadiz University, 11001 Cadiz, Spain

**Keywords:** caregivers, children and adolescents, food choice, nutritional status, type 1 diabetes

## Abstract

Parents play a key role in what their children eat. The Food Choice Questionnaire (FCQ) has been used elsewhere to assess the dietary motivations of parents of healthy children, but not for parents of children with chronic diseases such as type 1 diabetes (T1D). The aim of our research was to evaluate the associations between parental food choice motivations and the nutritional status and glycemic control of children with T1D. A cross-sectional observational study of children aged 5 to 16 years with T1D attending the Pediatric Endocrinology Unit of Puerta del Mar University Hospital in Cádiz (Spain) was performed. Demographic, anthropometric and clinical data, including glycated hemoglobin, were collected. The FCQ in Spanish was conducted to assess the eating behaviors of the main caregivers of children with T1D. Significance was established at the level of *p*-value < 0.05. In total, 85 children with T1D (female 56.5%, age 12.07 ± 2.93 years, HbA1c 7.29 ± 0.77%) were recruited. Of these children, 31.3% showed HbA1c levels of <7.0% and 44.9% had a TIR >70%. A significant positive correlation was found between Hb1Ac and “familiarity” (R: +0.233). Anthropometric measures (weight, BMI, skinfolds and body circumferences) showed significant positive correlations with “sensory appeal” and “price”. Parents’ eating behaviors influence the nutritional status of their children with T1D and their glycemic control of the disease.

## 1. Introduction

Type 1 diabetes (T1D) is the most common form of diabetes in young people [1]. It is a condition caused by autoimmune damage to the insulin-producing beta cells of the pancreatic islets, usually leading to severe endogenous insulin deficiency [2]. The estimated prevalence in Spain is between 1.1 and 1.4/1000 children under 15 years old [3]. The incidence in the pediatric population in Spain is estimated at 17.69 cases/100,000 inhabitants per year [4]. There is no cure for type 1 diabetes, so optimal treatment is required to enable the best possible quality of life and ensure the effective use of healthcare resources while minimizing the risk of complications [5]. In addition to intensive insulin therapy, an essential element of the treatment of T1D in children is the diet, in particular carbohydrate intake, which has an important effect on blood glucose levels and is associated with a reduction of HbA1c [6]. The nutritional strategy for children and adolescents is based on balanced, healthy meals, in which an appropriate ratio of carbohydrates to proteins and fats should be maintained [7].

However, childhood adiposity is a serious problem worldwide [8]. Healthy eating is a particularly critical factor influencing the weight of children [9], and a concern for young children with T1D who often have diets with excessive saturated fat and inadequate intake of vegetables and fruit [10]. A prevention of childhood adiposity study showed that controlling diet was more effective than interventions designed to increase the level of physical activity [11]. However, eating habits and child weight are difficult to modify directly [12].

Food preferences continue to change throughout a person’s life, and they are key determinants of food choices and therefore diet quality [13]. Parents are the most important environmental factors affecting the formation and maintenance of their children’s eating habits [14]. In the first years of life, children are totally dependent on adults for the provision of nutritious food [15]. During these early years, there are many barriers that parents face when providing a healthy diet for their children (expense, lack of time or knowledge, etc.). However, Patton and collaborators highlighted a unique challenge in the case of parents of children with T1D, namely parents’ desire not to limit their child’s diet or make their child “feel different” [16]. In particular, parental food choices determine the type and quality of food that a child consumes at home. Moreover, the consumer behaviors of parents, such as buying particular unhealthy foods, may determine their children’s diet as they act as gatekeepers [17]. Thus, parental feeding practices are potentially a good target for interventions to prevent unhealthy eating patterns [18], the development of obesogenic eating behaviors and excessive weight gain in young children and adolescents [19].

Despite family involvement being a vital component of optimal diabetes management throughout childhood and adolescence [20], to our knowledge, how parental food choices act as precursors to healthy feeding practices in T1D children has not yet been examined.

The aim of this study was to evaluate the association between parental food choice motives (“mood”, “health and natural content”, “sensory appeal”, “weight control”, “convenience”, “familiarity” and “price”) and the nutritional status and glycemic control of children with T1D.

## 2. Materials and Methods

### 2.1. Study Design and Participants

This cross-sectional observational study was based on a survey of the main caregivers of children 5 to 16 years of age, with a diagnosis of T1D with more than 1 year of evolution, treated as outpatients at the Pediatric Endocrinology Unit of Puerta del Mar University Hospital in Cádiz (Spain) from February to May 2022. Patients outside this age range or/and who had been diagnosed with T1D for less than 1 year were excluded. Children without sociodemographic, clinical and biochemical data were also excluded. Parents with a serious nutritional condition were not surveyed and their children were not part of the final sample. As the study involved human participants, it was reviewed and approved by the Ethics Committee of Research of Cadiz. Written informed consent to participate in this study was provided by the participants’ parents.

### 2.2. Procedures and Measures

Face to face interviews were conducted with the participants on the same day as their children’s medical consultation, after an explanations of how to fill in the questionnaire was provided by a trained investigator. Patients’ sociodemographic (sex, age, pubertal stage and ethnic group) and clinical data (insulin therapy, total daily insulin, HbA1c level and ambulatory glucose profile parameters) were collected from digital clinical records. Biochemical data from most recent last blood test were also included (GOT, GPT, venous HbA1c, total cholesterol, HDL, LDL and triglyceride levels). Anthropometric measurements were recorded for each individual by trained pediatric endocrinology physicians following standard operating procedures. For quality assurance, the same equipment was used to obtain anthropometric data from all the participating subjects. Height was measured using a portable stadiometer (ADE MZ10042) consisting of a vertical stand and an adjustable headpiece. Height was measured to the nearest 0.1 cm. Correct body posture was maintained while recording the standing height. Patient weight was measured with light clothing and without shoes using a Tanita model DC-430 S MA to the nearest 0.2 kg. Body mass index was calculated using the formula BMI = Kg/m^2^. Three skinfold measurements were taken using a Holtain caliper (accurate to 0.2 mm) from the following sites: triceps, subscapular and iliac crest. Waist circumference and mid-upper-arm circumference (MUAC) were also measured. Z-scores for age and gender for every anthropometric variable were also registered using Spanish reference values.

Using bioelectrical impedance (BIA) (Tanita model DC-430 S MA), the following parameters were obtained: fat-free mass, body fat mass and total body water. As recommended by the manufacturer, each participant stood on the scale barefoot in contact with the four electrodes for the feet, dressed in light clothing and with the arms separated from the trunk to prevent contact.

### 2.3. Food Choice Questionnaire (FCQ)

The Food Choice Questionnaire (FCQ) [21], in its Spanish version (FCQ-SP) [22], was use to analyze parental nutritional choices. The Food Choice Questionnaire (FCQ) is a 34-item questionnaire comprising seven different intrinsic and extrinsic food attributes (“mood”, “health and natural content”, “sensory appeal”, “weight control”, “convenience”, “familiarity” and “price”), which may motivate consumers when choosing foods. Each item allows the respondent to grade the relevance of the food choice on any given day using a 7-point scale (1 = not important to 7 = very important).

### 2.4. Statistical Analysis

All analyses were conducted using SPSS statistical software (version 25, IBM Corp, 2017). The descriptive analysis provided means and standard deviations for continuous variables and frequencies and percentages for categorical variables. Student’s *t*-test was performed to compare continuous variables and *χ*^2^ was used when proportions were compared.

Bivariate analysis using simple linear correlation were used to assess the relationships between the parent’s food choice scores (dependent variable) and several independent variables, such as age, HbA1c levels, carbohydrate consumed, TDI, height, weight, BMI, fat mass and percentage, skinfolds and circumferences.

The cut-off point for statistical significance used was *p* ≤ 0.05.

## 3. Results

Eighty-five patients with T1D were included (48 girls), with ages ranging from 5 to 16 years. The mean age of the children in the sample was 12.04 ± 2.93 years and 18 (22%) were pre-pubertal. The mean age at diabetes diagnosis was 6.44 ± 3.28 years. Fifty-seven of them (67.1%) were receiving multiple daily injection (MDI) therapy and 28 (32.9%) were being treated with a closed-loop system. The clinical and biochemical characteristics of the patients included are summarized in Table 1.

### 3.1. Nutritional Status and Metabolic Control of T1D

The mean participant HbA1c level was 7.29 ± 0.77%. Among all the study participants, 25 (31.3%) showed HbA1c levels of <7% and 35 (44.9%) showed a value in the range of >70%. Table 2 summarizes the anthropometrical characterization. Normal weight was recorded for 70.6% of patients in the study group, 17.64% of children were overweight (BMI > 1 z-score) and 11.8% presented with obesity (BMI > 2 z-score). Skinfolds, waist and mid-upper-arm circumference z-scores were also analyzed in this study. Patients showed triceps and subscapular skinfold z-score means of 0.77 ± 0.98 and 0.61 ± 1.0, respectively. In the present sample, 9.4% (n = 8) of children had a triceps skinfold z-score > 2 and only 1.17% (n = 1) had a subscapular skinfold z-score > 2. The sample children showed a waist circumference z-score mean of 1.49 ± 2.96. Only 2.35% (n = 2) of patients had a waist circumference z-score < −1. However, 21.2% (n = 18) of children had a waist circumference z-score > 2. Similar results were found for the MUAC z-scores: z-score mean of 1.18 ± 1.39; 2.35% (n = 2) of children had a MUAC z-score < −1 and 21.2 (n = 18) had a MUAC z-score > 2.

The bioimpedanciometry results showed a mean fat-free mass of 33.78 ± 10.42 kg, a mean body fat mass of 14.94 ±9.99 kg and 26.39 ± 11.48% and a mean total body water of 25.58 ± 6.30%.

### 3.2. Parental Food Choices

The main caregivers were mostly women (95.3%), and were mothers in the majority of cases (94.1%). The average age of participating parents was 43.9 ± 7.21 years. Of these, 24.7% stated that their highest educational level was middle school or lower, 12.9% stated that they had attended high school, 35.3% reported a technical degree as their highest educational level and 27.1% of the parents were classified as graduates or higher. Further sociodemographic data of the parents are given in Table 3.

A correlation study was performed using the scores from the FCQ food attributes and other independent variables. These data are shown in Figure 1.

“Sensory appeal” was the food attribute with the highest score (31.8%). A positive correlation was found between “sensory appeal” and the age of children (r = 0.279, *p* = 0.009) and also with children’s pubertal stage (r = 0.239, *p* = 0.029). This food attribute also showed a positive correlation with insulin needs (r = 0.302, *p* = 0.009).

Weakly positive associations were found between “familiarity” and both levels of HbA1c measured: capillary HbA1c (r = 0.233, *p* = 0.032) and venous HbA1c (r = 0.219, *p* = 0.05).

There were positive correlations between “convenience” and the percentage of prandial insulin (r = 0.213, *p* = 0.044) and the carbohydrate consumed (CH/day) (r = 0.253, *p* = 0.047); (CH/kg) (r = 0.271, *p* = 0.033); (kcal/CH/(kg) (r = 0.271, *p* = 0.033).

Further positive correlations were found between “sensory appeal” and anthropometric measures: percentage of body fat mass (r = 0.225, *p* = 0.045), weight (r = 0.246, *p* = 0.022), triceps skinfold (r = 0.251, *p* = 0.022), subscapular skinfold (r = 0.224, *p* = 0.042), iliac skinfold (r = 0.3, *p* = 0.006), waist circumference (r = 0.239, *p* = 0.03) and MUAC (r = 0.261, *p* = 0.018).

Additional positive associations between “weight control” and anthropometric measures were also found: percentage of body fat mass (r = 0.309, *p* = 0.005), weight z-score (r = 0.270, *p* = 0.012), BMI z-score (r = 0.276, *p* = 0.01), subscapular skinfold (r = 0.230, *p* = 0.037), waist circumference (r = 0.231, *p* = 0.036), waist circumference z-score (r = 0.233, *p* = 0.034), MUAC (r = 0.243, *p* = 0.028) and MUAC z-score (r = 0.279, *p* = 0.011).

“Price” showed strongest positive correlations with anthropometric measures: percentage of body fat mass (r = 0.365, *p* = 0.001) and kilograms of body fat mass (r= 0.337, *p* = 0.002), weight z-score (r = 0.287, *p* = 0.007), BMI z-score (r = 0.301, *p* = 0.005), triceps skinfold (r = 0.341, *p* = 0.002), triceps skinfold z-score (r = 0.324, *p* = 0.003), subscapular skinfold (r = 0.306, *p* = 0.005), subscapular skinfold z-score (r = 0.288, *p* = 0.009), iliac skinfold (r = 0.343, *p* = 0.002), waist circumference (r = 0.306, *p* = 0.005), MUAC (r = 0.24, *p* = 0.03) and MUAC z-score (r = 0.281, *p* = 0.011).

Finally, positive associations between parental food choices and their sociodemographic characteristics were also seen. There was a positive correlation between “convenience” and education level (r = 0.269, *p* = 0.012) and between “familiarity” and age (r = 0.252, *p* = 0.02).

## 4. Discussion

The results of this study revealed a positive correlation between the anthropometric measures of children (weight, BMI, skinfolds, body circumferences and percentage of fat mass) and questionnaire responses related to food attributes such as “sensory appeal”. Namely, there was an increase in weight gain among children whose parents attached more importance to the taste, smell and texture of the food they consume. Parents, as the children’s role models, are a major environmental factor affecting the formation of their children’s eating habits [11]. A similar study conducted in children aged 6 to 12 years from seven European countries suggested that parental consumer attitudes are associated with their children’s food intake but not with their taste preferences [17]. However, the associations between parental taste preferences and their children’s food intake were not analyzed.

Price was another factor positively linked with higher anthropometric parameters in our study. Family affluence, considered a marker of socioeconomic status [23], has been associated with the development of healthy food choices in children [24,25]. Similarly, more frequent consumption of fresh vegetables and fruits has been associated with higher socioeconomic status in children and young people in Norway [26], Canada [27], Greece [28], Spain [29], the United Kingdom [30] and Iran [31].

In high-income countries, childhood socioeconomic status is inversely associated with weight among children and adolescents [32], with parents’ education yielding the most consistent association [33]. However, according to our results, when the educational level of parents was analyzed, a direct positive correlation with the “convenience” factor was observed. This means that parents who were classified as graduates or higher showed higher scores for items such as “is easy to prepare”, “can be cooked very simply” or “takes no time to prepare”. Previous studies showed that participants with lower education and lower socioeconomic status spent more time preparing food daily than the participants with the highest education and highest socioeconomic status [34]. In our investigation, this “convenience” attribute also presented a direct positive correlation with the percentage of prandial insulin and the amount of carbohydrate consumed (measured in: HC/day, HC/kg and kcal/HC/kg). Both factors could be related to a higher consumption of precooked and processed food, instead of raw or fresh food.

HbA1c levels presented a positive correlation with familiarity. This means that the children of parents who preferred foods similar to those they consumed during their childhood and adolescence had higher HbA1c levels. Previous studies have shown that neophobia (avoidance of unfamiliar food) is familial and is associated with less healthy food choices [35]. Those unhealthy food choices made by parents and children could be related to a lower likelihood of achieving optimal glucose control parameters in T1D. However, these results could also have another explanation. Parents with fear of hypoglycemia prefer to choose familiar foods because they know that their children are also familiar with them and will not reject them. This lower variety in the diets of parents of children with T1D has already been observed in parents of children with other chronic diseases during childhood, such as phenylketonuria [36], and it seems to be related to parental pressure surrounding the central message that metabolic parameters should always be maintained within target range, avoiding making mistakes with the introduction of new foods [35].

Finally, children’s age was another critical parameter in our sample, with a direct correlation with “sensory appeal”. In line with previous research, adolescents have a greater tendency to consume “junk” (energy-dense/low-nutrient) food [37,38], and this consumption appears to increase through adolescence [23]. Consistent with other research, when the FCQ was applied to the parents of toddlers [15] and preschool children [39], higher scores for the motive of “health” were seen. This supports the idea that as children grow older, their decision-making capacity about the foods they consume increases, as well as their influence on the family “shopping list”.

The main strength of this study is that the questionnaire used in the research was validated and reliability tested and has been widely used in other studies. Additionally, the study is particularly important because there is a lack of this type of study on children with T1D.

An important limitation of this study involves its cross-sectional design, which complicated causal interpretation of the results. Although the size of the sample included was acceptable, a more representative study sample might have given different results. Another limitation is that none of the items showed a strong correlation with any variable, with 0.365 being the highest R value of all the correlations analyzed (a positive correlation between “price” and the percentage of body fat mass). Therefore, although there were statistically significant positive correlations, their clinical relevance could be limited.

## 5. Conclusions

In summary, our results suggest that when parental food choices are driven by the considerations of “price” and “sensory appeal”, this seems to have a negative impact on the nutritional status and body composition of their children with T1D, whereas choosing foods based on “convenience” and “familiarity” negatively impacts the children’s glucose profiles in terms of higher prandial insulin requirements and higher HbA1c. Furthermore, this study highlights the importance of educating parents continuously about their role and about the importance of managing their own eating habits to educate their children about healthy eating habits.

## Figures and Tables

**Figure 1 foods-12-01969-f001:**
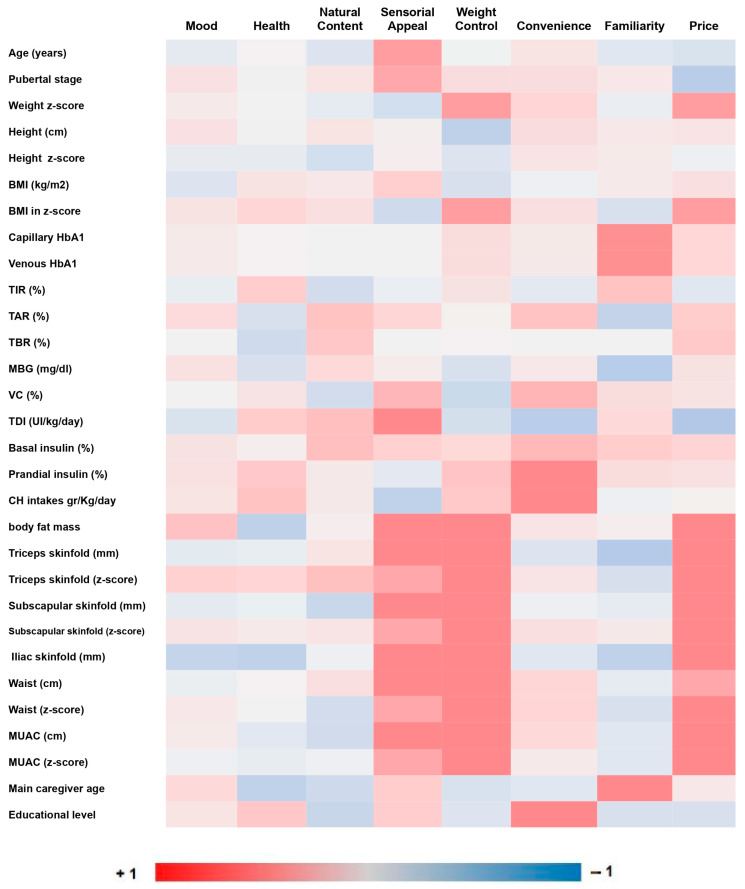
Correlation analysis between FCQ food attributes and other independent variables. Abbreviations: BMI: body mass index; HbA1c: capillary glycated hemoglobin; TIR: time in range; TAR: time above range; TBR: time below range; MBG: mean blood glucose; VC: variation coefficient; TDI: total dose insulin; MUAC: mid-upper-arm circumference measurement.

**Table 1 foods-12-01969-t001:** Clinical, biochemical and metabolic characteristics of children with T1D (*N* = 85).

Clinical Characteristics	All	Female	Male	*p*-Value
N (%)	85	48 (56.5)	37 (45.5)	-
Age in years, mean (SD)	12.35 (3.07)	11.8 (3.3)	12.6 (2.8)	0.254
Ethnic group				
Caucasian, n (%)	79 (92.9)	45 (93.7)	33 (89.2)	0.729
North African, n (%)	4 (4.8)	2 (4.1)	2 (5.4)	
Others, n (%)	2 (2.4)	1 (2)	2 (5.4)	
Diabetes duration in years, mean (SD)	6.5 (2.1)	6.7 (1.9)	7.3 (2.1)	0.321
DKA at diagnosis, n (%)	32 (37.6)			
Insulin therapy				
Multiple doses of insulin, n (%)	57 (67.1)	23 (62.2)	34 (70.8)	0.260
Closed-loop system, n (%)	28 (32.9)	14 (37.8)	14 (29.2)	
Associated diseases				
Hypothyroidism, n (%)	4 (4.7)	1 (2.7)	3 (6.3)	0.957
Celiac disease, n (%)	6 (7.1)	2 (5.4)	4 (8.3)	
Graves’s disease, n (%)	1 (1.2)	0 (0)	1 (2.1)	
Hypercholesterolemia, n (%)	2 (2.4)	2 (5.4)	0 (0)	
Hypertension, n (%)	1 (1.2)	1 (2.7)	0 (0)	
Diabetes metabolic characteristics				
HbA1c (%), mean (S)	7.3 (0.7)	7.34 (0.8)	7.24 (0.8)	0.765
AGP parameters, mean (S)				
Time in range (%)	65.49 (15.8)	65.3 (16.8)	66.3 (14.7)	0.297
Time above range (%)	31.08 (16.4)	31.7 (17.2)	29.8 (15.2)	0.260
Time below range (%)	2.8 (2.9)	3 (2.5)	2.8 (2.6)	0.242
Mean blood glucose (mg/dL)	165.3 (30.4)	164.6 (31.8)	165.8 (29.3)	0.381
Variation coefficient (%)	34.9 (5.6)	35.1 (5.3)	34.7 (5.9)	0.523
Total dose insulin in UI/kg/day, mean (S)	0.9 (0.2)	42.8 (22.9)	51.9 (20.8)	0.673
Basal insulin (%), mean (S)	43 (11.5)	43.8 (11.1)	42.6 (12)	0.446
Prandial insulin (%), mean (S)	57.1 (11.3)	57.2 (10.5)	57.4 (12)	0.441
CH intakes in gr/Kg/day, mean (S)	1.89 (1.53)	2.3 (1.7)	1.7 (1.3)	0.183
Biochemical characteristics				
AST (UI/L)	19.93 (6.85)	22.8 (7.4)	19.6 (5.4)	0.123
ALT (UI/L)	17.06 (12.08)	17.2 (6.1)	17 (8.1)	0.457
HbA1 (venous)	7.4 (1.28)	7.3 (1.2)	7.4 (1.3)	0.466
Total cholesterol (mg(dL)	167.7 25.6)	169.1(28.4)	167.7 (22.2)	0.181
HDL-c (mg/dL)	62.02 (12.44)	61.3 (12.1)	63 (12.5)	0.242
LDL-c (mg/dL)	93.4 (22.85)	95.4 (26.5)	92.6 (19.2)	0.335
Triglycerides (mg/dL)	66.2 (26.89)	61.6 (22)	69.1 (29)	0.165

Abbreviations: AGP: ambulatory glucose parameters; HbA1c: capillary glycated hemoglobin; AST: aspartate aminotransferase; ALT: alanine aminotransferase; HDL-c: high-density lipoprotein cholesterol; LDL-c: low-density lipoprotein cholesterol.

**Table 2 foods-12-01969-t002:** Anthropometric parameters of children with T1D (*N* = 85).

Anthropometric Parameters	All	Female	Male	*p*-Value
Weight (kg), mean (SD)	51.(16.9)	47.6 (18.1)	52.6 (15.7)	0.146
Weight z-score, mean (SD)	0.5 (0.9)	0.28 (0.9)	0.58 (1)	0.256
Height (cm), mean (SD)	150.3 (22.4)	151.2 (19.1)	149.3 (24.9)	0.484
Height in z-score, mean (SD)	0.16 (1.04)	0.25 (0.9)	0.8 (1.1)	0.382
BMI (kg/m^2^), mean (SD)	22 (7.5)	20 (3.8)	23.68 (9.2)	0.112
BMI in z-score, mean (SD)	0.48 (0.99)	0.15 (0.9)	0.7 (1)	0.213
Skinfolds, mean (SD)				
Triceps (mm)	17.3 (6.1)	14.6 (5.7)	19.3 (5.6)	0.001
Triceps (z-score)	0.77 (0.9)	0.71 (1)	0.83 (1)	0.712
Subscapular (mm)	12.05 (5.5)	10.1 (4.7)	13.5 (5.8)	0.012
Subscapular (z-score)	0.61 (1.0)	0.56 (0.8)	0.68 (1)	0.539
Iliac (mm)	10.50 (5.5)	8.46 (5.6)	11.8 (5.6)	0.002
Body circumferences				
Waist (cm)	72.7 (10.8)	70.3 (11.6)	72.4 (10.1)	0.125
Waist (z-score)	1.5 (2.9)	1.64 (1.3)	1.9 (1.6)	0.231
MUAC (cm)	24.8 (4.6)	23.9 (3.9)	25.1 (5.5)	0.250
MUAC (z-score)	1.2 (1.4)	1.12 (1.5)	1.32 (1.3)	0.325
Tanner stage, n (%)				
I	18 (22.0)	6 (16.2)	12 (63.2)	0.01
II–IV	25 (29.4)	12 (32.4)	13 (27)	
V	39 (47.5)	30 (62.5)	9 (26.5)	
SBP (mmHg), mean (SD)	112.6 (10.6)	111 (11.3)	113.8 (10.2)	0.144
SBP SD, mean (SD)	0.58 (0.8)	0.37 (0.9)	0.76 (0.7)	0.456
DBP (mmHg), mean (SD)	72.7 (6.6)	70.4 (7)	74.4 (6)	0.152
DBP SD, mean (SD)	0.86 (0.5)	0.71 (0.66)	0.98 (0.5)	0.368

Abbreviations: BMI: body mass index; MUAC: mid-upper-arm circumference measurement; SBP: systolic blood pressure; DBP: diastolic blood pressure.

**Table 3 foods-12-01969-t003:** Characteristics of participating caregivers of children with T1D (*N* = 85).

Characteristics of Caregivers	
Age in years, mean (SD)	43.9 (7.2)
Female, n (%)	81 (95.3)
Relationship, n (%)	
Mother	80 (94.1)
Father	4 (4.7)
Grandmother	1 (1.2)
Smoker, n (%)	19 (22.4)
Education level	
Unfinished middle school, n (%)	1 (1.2)
Middle school, n (%)	20 (23.5
High school, n (%)	11 (12.9)
Technical degree, n (%)	30 (35.3)
University, n (%)	23 (27.1)
Employed, n (%)	48 (56.5)
Diseases	
Hypothyroidism, n (%)	6 (7.1)
Celiac disease, n (%)	2 (2.4)
Graves’s disease, n (%)	3 (3.5)
Hypercholesterolemia, n (%)	1 (1.2)
Hypertension, n (%)	2 (2.4)
T1D, n (%)	3 (3.5)
Obesity, n (%)	7 (8.2)
Cancer, n (%)	1 (1.2)

Abbreviations: BMI: body mass index; MUAC: mid-upper-arm circumference measurement; SBP: systolic blood pressure; DBP: diastolic blood pressure.

## Data Availability

Data is contained within the article.

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
