# Peer review of "Impact of Parental Food Choices on Nutritional and Metabolic Status of Children with Type 1 Diabetes"

_foods, 2023, doi:10.3390/foods12101969_

Round 1

Reviewer 1 Report

The authors investigate the impact of parental food choices on children with T1D.

In the abstract T1D should be defined in the first instance, then the correct abbreviation (ie not DT1) used throughout.

The data mentioned in lines 161+ should be clearly presented as a table or figure. The correlations should be made into a figure. 

Minor

Remove "of this syndrome" (line 31)

Explain why diet is critical for T1D (line 37)

Line 39-44 is not relevant. T1D are typically lean children. Especially as BM! (table 2) is 22

What is the hypothesis or aim of this study

Line 79 what is Tanner stage?

The way the tables are presented is confusing. For example it was not clear what the values for female meant (ie the % sign should be after the female), also 

The data in lines 161+-this refers to the parents food choices-how does this impact the child? A parent may choose something completely different for themselves compared to their T1D child. The correlations are between their childrens anthropometric measures of their children to their parents food choices (not the childrens food)

Author Response

Dear reviewer,  

Thanks for your review and contribution to improve our work. 

Find  the  responses to each comment and changes suggested below: 

 1.-  In the abstract T1D should be defined in the first instance, then the correct abbreviation (ie not DT1) used throughout.

Type 1 diabetes was defined in the first place in the abstract and the abbreviation was also corrected.

2.- mThe data mentioned in lines 161+ should be clearly presented as a table or figure. The correlations should be made into a figure.

Correlation analysis between FCQ food attributes and other independent variables have explained as a new figure in figure 1.

3.- Remove "of this syndrome" (line 31)

Done.

4.- Explain why diet is critical for T1D (line 37)

The explanation of the importance of diet in children with type 1 diabetes has been included in the manuscript (lines 36-37), in accordance with the reviewer suggestions.

5.- Line 39-44 is not relevant. T1D are typically lean children. Especially as BM! (table 2) is 22.

Weight control in type 1 diabetes patients is important due to increased cardiovascular risk even in lean patients. Because of this reason, we consider this paragraph relevant.

6.- What is the hypothesis or aim of this study?

The aim of this study is described in lines 68 to 70: “The aim of this study is to evaluate the association between nutritional status and glycemic control of children with DT1, with parental food choice motives (mood, health and natural content, sensory appeal, weight control, convenience, familiarity and price)”

7.- Line 79 what is Tanner stage?

To simplify the concept and clarify the meaning of “Tanner stage”, we have changed “Tanner stage” to “pubertal stage”.

8.- The way the tables are presented is confusing. For example, it was not clear what the values for female meant (ie the % sign should be after the female), also.

In order to make the tables easier to understand, they have been modified according to the reviewer suggestions.

9.- The data in lines 161+-this refers to the parents’ food choices-how does this impact the child?

A parent may choose something completely different for themselves compared to their T1D child. The correlations are between their children anthropometric measures of their children to their parents’ food choices (not the children food) Parent food preferences modify their food choices, which directly influence not only their own eating habits, but also those of their children. This is because it is parents who, through their choices (weekly shopping, menus, etc.) determine the type and quality of their childrens diet.

Reviewer 2 Report

I appreciate the opportunity to review the article "Impact of parental food choices on nutritional and metabolic status of children with type 1 diabetes."

The topic is relevant and fits the journal scope. The manuscript is well-written and has a good sequence. However, there are significant comments that need to be addressed for a new evaluation:

Introduction

- Line 41. Authors state, "studies have shown…." However, there is just one reference cited. 

- Lines 45-46. These lines are not right in this part of the paragraph. I suggest removing them. 

- Lines 45 to 56. I suggest considering the parents' food choices as the main topic. These are not the same as parents' food preferences. Food preferences influence food choices as other variables such as food price, time, knowledge, etc. 

- I suggest reviewing other studies to incorporate into this introduction, such as: 

Patton, S. R., Clements, M. A., George, K., & Goggin, K. (2016). "I Don't Want Them to Feel Different": A Mixed Methods Study of Parents' Beliefs and Dietary Management Strategies for Their Young Children with Type 1 Diabetes Mellitus. Journal of the Academy of Nutrition and Dietetics116(2), 272–282.

Seckold R, Howley P, King BR, et al. Dietary intake and eating patterns of young children with type 1 diabetes achieving glycemic targets BMJ Open Diabetes Research and Care 2019;7:e000663. doi: 10.1136/bmjdrc-2019-000663

- Line 58. It says: "…throughout childhood and adolescence [20]. To our knowledge…" It should be: throughout childhood and adolescence [20], to our knowledge…"

Material and methods

- How many patients does the unit attend? I would like to see what a sample size of 85 participants could mean. 

- I suggest adding references about anthropometric protocol. 

- Statistical analysis. Some analysis needs to be included here. The authors state that they conducted a simple linear regression; however, the results section shows a correlation analysis. Please, clarify.

- Why was age an independent variable?

- Were there differences between females and males? 

- This section lacks how variables were analyzed. For example, the z score for skinfolds. This analysis appears in line 135 in the result section. 

Results

- Where are the results of the linear regression?

- After comparing females and males, the tables could change. 

- The discussion section should be improved according to the new result section. 

Author Response

Dear reviewer,  

Thanks for your review and contribution to improve our work. Find below responses to your   comments

Introduction

X Line 41. Authors state, "studies have shown…." However, there is just one reference cited. 

Suggestion was accepted and the sentence was correctly phrased.

Lines 45 to 56. I suggest considering the parents' food choices as the main topic. These are not the same as parents' food preferences. Food preferences influence food choices as other variables such as food price, time, knowledge, etc

The authors agree with the reviewers suggestion that parents’ food; should be the main topic in the manuscript and that is why it is already stated in the title of the manuscript. As pointed out in line 57, food preferences are determinants of parental food choices, which are also influenced by parental attitudes and knowledge (line 61). These choices modify the type and quality of childrens diets and, according to our hypothesis, their nutritional and metabolic status.
To clarify that food choices is the main topic of the manuscript, the definition of Food Choice. Questionnaire (lines 119-120) has been rephrased.

I suggest reviewing other studies to incorporate into this introduction, such as: 
Patton, S. R., Clements, M. A., George, K., & Goggin, K. (2016). "I Don't Want Them to Feel Different: A Mixed Methods Study of Parents' Beliefs and Dietary Management Strategies for Their Young Children with Type 1 Diabetes Mellitus. Journal of the Academy of Nutrition and Dietetics, 116(2), 272–282.

Seckold R, Howley P, King BR, et al. Dietary intake and eating patterns of young children with type 1 diabetes achieving glycemic targets BMJ Open Diabetes Research and Care 2019;7:e000663. doi: 10.1136/bmjdrc-2019-000663
Both suggested articles have been analyzed and incorporated into the introduction of this manuscript
given the relevance of the topic.

Line 58. It says: …throughout childhood and adolescence [20]. To our knowledge… It should be: throughout childhood and adolescence [20], to our knowledge…;

Suggestion was accepted and the sentence was correctly phrased.

Material and methods

How many patients does the unit attend? I would like to see what a sample size of 85 participants could mean. 

Our unit attends 115 children and adolescents with type1 diabetes and our sample on this study is considered representative in age and sex of our unit patients. No included patients were mainly underage.

I suggest adding references about anthropometric protocol. 

Anthropometric explanation was added as suggested in line 100.

Statistical analysis. Some analysis needs to be included here. The authors state that they conducted a simple linear regression; however, the results section shows a correlation analysis. Please, clarify.

We didn’t include linear regression in this study. We perform correlation analysis. The sentence has been modified to clarify.

Why was age an independent variable?
We consider dependent variable then score in FCQ and independent variable which could affect or causative of change on FCQ as age etc.

Were there differences between females and males? 
Analysis in between sexes were performed and no relevant differences were found, unless in pubertal stage. Analysis have been included in table 1 and 2.

X This section lacks how variables were analyzed. For example, the z score for skinfolds. This analysis appears in line 135 in the result section.

Z-score for skinfolds was determined by Spanish reference tables. This data is included in methods.

Results

Where are the results of the linear regression?
No linear regression was performed, only correlation analysis. Clarification was added on methods.

After comparing females and males, the tables could change

Done.

The discussion section should be improved according to the new result section.

Little changes in discussion have been done in order to improve the new results.

Round 2

Reviewer 2 Report

Thanks to the authors for this new version.

I do not have any comment for this version.